# What is associated with reported acute respiratory infection in children under 5 and PCV vaccination in children aged 1–36 months in Malawi? A secondary data analysis using the Malawi 2014 MICS survey

**Justine Gosling[1,2], Tim Colbourn**[1] *

**1** UCL Institute for Global Health, London, United Kingdom, **2** World Health Organization Regional Office for Europe, Copenhagen, Denmark

* t.colbourn@ucl.ac.uk

## Abstract

### Introduction

Acute respiratory illness (ARI) is a leading cause of mortality in children under 5 (CU5) in Malawi and can be prevented with 3-dose pneumococcal conjugate vaccine (PCV). There has been no national study in Malawi that seeks to associate social economic factors leading to PCV vaccine uptake and reported acute respiratory infections (RARI). The objectives of our study were to do this.

### Methods

We conducted a cross-sectional analysis of secondary data from the 2014 UNICEF Malawi Multiple Indicator Cluster Survey to construct mutlivariable logistic regression models for independent associations with PCV 1/2/3 immunisation and RARI.

### Results

56% of CU5 in Malawi RARI in the 2 week recall period of the survey. Independent associations with reduced odds of RARI were central region living (OR 0.82, 95%CI (0.71–0.93)) middle (OR 0.84, (0.73–0.97)) fourth (OR 0.79, (0.68–0.92)) and richest wealth quintiles (OR 0.73, (0.60–0.88)). Using straw/shrubs for fuel was associated with increased RARI (OR 3.13, (1.00–9.79)). Among 1–36 month olds, in 2014, 93.3% received PCV1, 86.8% PCV2 and 77.0% PCV3. Between 2011–2014, the average age in months for a child to receive PCV1/2/3 reduced by 26.6 for PCV1, 26.4 for PCV2, and 26.1 for PCV 3. Independent predicators for increased odds of all 3 PCV doses, relative to 0–5 age group, were age group 6–11 (OR 21.8, (18.2–26.1) 12–23 (OR 27.5, (23.5–32.2) 24–36 months (OR 9.09, (7.89–10.5), mothers having a secondary (OR 1.52, (1.25–1.84)) or higher education (OR 2.68, (1.43–5.04) when compared to no education, and children in the middle (OR 1.24, (1.07–1.43)) fourth (OR 1.27, (1.09–1.48)) richest (OR 1.54, (1.27–1.88)) wealth quintiles

**Data Availability Statement:** We used data from the 2014 UNICEF Malawi Multiple Indicator Cluster Survey (https://mics.unicef.org/surveys), which is

freely available online. From this we created a
minimal data set which we include in the
Supporting information files. We also include, as
Supporting information, Stata code used to create
the minimal data set from the MICS 2014 datasets
and reproduce the analyses in the paper.

**Funding:** The author(s) received no specific
funding for this work.

**Competing interests:** The authors have declared
that no competing interests exist.

**Abbreviations:** ARI, Acute respiratory illness;
children, CU5, Children under 5; CI, Confidence
intervals; MICS, Multiple Indicator Cluster Survey;
ORs, Odds ratios; PCV, Pneumococcal conjugate
vaccine; RARI, Reported acute respiratory
infections; SDG, Sustainable Development Goal;
SEAs, Sample enumeration areas; WHO, World
Health Organisation.

relative to the lowest. Children living with 4–6 other children was independently associated
with reduced odds of receiving all 3 PCV doses (OR 0.56, (0.33–0.96).

## Conclusion

We report nationally representative social economic associations with RARI and PCV vac-
cine uptake and coverage estimates. We found reductions in the average age a child
receives all 3 PCV vaccine doses between 2011–2014.

## Introduction

In 2019, a child less than 5 years old died from pneumonia every 39 seconds [1]. Although
numbers are declining, pneumonia persistently remains the leading infectious cause of death
for children under 5 (CU5) globally, killing around 880,000 children in 2019 [1]. Current
trends estimate that 735,000 children will die due to pneumonia in 2030 despite the Sustain-
able Development Goal (SDG) target 3.2,"ending preventable child deaths" [2].

Pneumonia diagnosis is strongly associated with a child's socio-economic status. Evidence
from Malawi demonstrates that rural dwelling [3, 4] malnourishment [5, 6] low income [6]
and increasing household density [6, 7] are factors that predispose children to pneumonia.

Almost all pneumonia deaths are preventable through vaccination, early diagnosis and low
cost treatment and care [8]. First licenced in 2000, the pneumococcal conjugate vaccine (PCV)
is a safe and efficacious vaccine to Streptococcus bacterial infection [9]. PCV13 inhibits naso-
pharyngeal carriage reducing transmission, allowing for the herd effect [10]. In 2007 the
World Health Organisation (WHO) published guidelines advising PCV be added to national
immunization programmes as a key strategy for rapidly reducing deaths from pneumonia
[11]. Distribution of Thirteen-valent PCV in Malawi began on November 12th 2011 and is
offered to infants in 3 doses at 6, 10 and 14 weeks of age as part of routine immunization [12].
The introduction coincided with catch up vaccinations with infants over 12 months receiving
3 doses at 1-month intervals [10]. The vaccination is free of charge and studies have shown it
to reduce cases of very severe pneumonia in children by 65% in both hospital and community
cases in Malawi [7].

Malawian coverage estimates for all 3 PCV doses by 12 months old from data collected
between 2012–2014 vary from 86% [13] 76% [7] increasing to 89.2% in 2015 [14] all just below
the target for full immunisation set at >90% [15]. Munthali [16] stated that in Malawi it is con-
sidered the responsibility of a child's mother to ensure they are vaccinated. The evidence is
inconsistent on whether maternal education level impacts PCV uptake with region specific
studies reporting no relationship [6] and others showing lower maternal education levels result
in lower children's vaccination rates [16–18]. Higher vaccine coverage has been reported in
large households [17] and in urban areas, likely due to greater access to health services [16].
Although pre PCV, Austin et al [18], demonstrated lower income households to be associated
with incomplete vaccination in Malawi, which is consistent with reports that the poorest in
society have less access to public health services [19]. No association has been found between
sex and uptake of other childhood vaccines [20]. Again, not specific to PCV, Abebe et al [20],
found regional vaccination coverage discrepancies with lower coverage in the northern region.
Probably owing to its fairly recent introduction in 2011, there are no studies investigating asso-
ciations with PCV vaccine uptake nationally, only regional studies.

Currently, rather than specify pneumonia, which is difficult to accurately diagnose based
on symptoms in the field [21], UNICEF [22] promotes the term "Acute respiratory illness"

(ARI) instead, an umbrella term for respiratory illness capturing pneumonia, comprising 2 symptoms: a cough and fast or difficulty in breathing. Based on this recommendation and because a survey is unable to diagnose pneumonia, this study will refer to reported acute respiratory illness (RARI) as our dependent variable definition encompassing the symptoms of pneumonia. ARI is widely used in research to define pneumonia in non-clinical settings [23].

Existing evidence suggests that rural Malawian children of lower social economic status are more likely to have RARI and less likely to be fully vaccinated with regional differences. To justify this study and research question, to date, there has been no study in Malawi that is generalisable nationally, that seeks to understand social economic factors associated with 1) PCV vaccine uptake and 2) RARI at household level. The existing evidence is limited to specific areas that is not generalisable or does not measure PCV since it was only introduced in November 2011. This study will be useful to understand the drivers of PCV vaccine uptake and social economic associations with RARI whilst estimating PCV vaccination coverage for 2014. Therefore, this study asks the question: What is associated with RARI in children under 5 and PCV vaccination in 1–36 month olds in Malawi?

**Primary objectives of this study are to establish:**

1. National, social economic associations with RARI in children under 5.

2. National, social economic associations with PCV uptake in children aged 1–36 months.

## Methods

### Research design

This study is a cross-sectional analysis of secondary data, utilising publicly available data from the 2014 UNICEF Malawi MICS, a country wide, 4 yearly household survey which monitors indicators of the lives and health of women and children in low income countries [24]. Anyone can access the data sets, however, researchers need to register with the MICS website to gain access. Individual informed consent was obtained at point of contact by the interviewers. The interviewer introduced themselves to the participant explaining that they were from the national statistics office and conducting a survey about the situation of children, families and households and asked for their consent to answer the questions [24]. Participants were reassured that information collected would remain confidential and anonymous [24].

### Sampling

Within Malawi, each district was stratified into urban and rural strata, yielding 56 sampling strata across the country. From each strata a specified number of census enumeration areas were independently selected systematically, with probability proportional to size. This process resulted in 1,140 sample enumeration areas (SEAs) and 28,479 households to be included in the 2014 survey. To form a sampling frame, a household listing was conducted within all the SEAs and a sample of 30 households per urban cluster and 33 per rural cluster systematically drawn. In this survey, a cluster is either an SEA or a segment of an SEA. One cluster had to be excluded as flooding prevented access at the time of the survey. All permanent residents aged between 15–49 were eligible for interview.

### Survey questionnaire

The MICS survey comprises 4 questionnaires. This study utilises the child questionnaire and a number of variables from the household questionnaire. The child questionnaire was administered to the caregivers of all residing children under 5. The English version was customised

and translated into Chichewa and Tumbuka and was pre-tested with appropriate modifications made to suit the survey population [24].

## Data collection

The survey was carried out between December 2013 and April 2014. Analysis of PCV can only be measured for 1–36 month olds as per available data from the MICS database and vaccine age criteria (there is no vaccination data on children aged 37–59 months). This age group permits children born after the introduction of the PCV in November 2011 up to the survey end date in April 2014.

## Ethical approval

Ethical approval was not needed as MICS data is anonymised.

## Measures

Field workers received 28 days of training to carry out the survey prior to implementation which included teachings on interviewing techniques, questionnaire contents and was finalised with mock interviews [24]. Data was collected by 32 teams comprising of 4 interviewers, a driver, editor, 1 measurer and a supervisor [24]. In addition to conducting questionnaires, fieldworkers measured the weights and heights of all CU5. Data was recorded using CSPro software, Version 5.0 and stored on 30 desktop computers. To reduce error, all questionnaires were double-entered and internally checked for consistency. Data processing ran simultaneously with data collection and was analysed using StataCorp [25].

## Study variables

**Dependent variables.**

1. RARI of CU5. Recorded if the caregiver answered "yes" when asked if the child had experienced "Difficulty in breathing with cough" in the last 2 weeks.

2. PCV1/2/3 vaccination status of children aged 1–36 months was obtained from vaccination record cards and, the absence of a vaccination card, caregiver recall.

**Independent variables.** We selected variables widely used in previous studies (see introduction) that aligned with our objectives.

*From the children's survey.* Age in months, age in 6 month categories, Urban/rural living, geographical region, sex, mothers reported education level in 5 categories, wealth index in 5 categories and the child's weight for height Z scores according to WHO standards.

The Weight for height Z score is widely accepted as the current, most representative and simple measure of malnutrition and its use is advised by the WHO [26]. Weight and height measures were collected and Z scores retrospectively calculated according to WHO standards and added to the data set [24]. Z scores express the anthropometric value as a number of standard deviations above or below the reference mean value and are independent of sex. This enables comparisons of a child's growth and malnutrition status across age groups to be made [26].

The wealth index is constructed of components including the ownership of consumer goods such as a television, refrigerator, motorcycle/scooter and the characteristics of the dwelling such as access to water and sanitation facilities, to generate scores. Scores are generated for the total sample and then stratified for urban and rural areas. These 2 strata scores are then

regressed on the score for the total sample to calculate the combined scores for the total population to minimise urban bias in the variable. A wealth score can then be calculated for each household which is ranked into one of 5 equal quintiles from lowest (poorest) to highest (richest). The wealth index is only applicable for the data set it is formed from [27].

*From the household survey.* Number of people residing in the house, number of children under 5 residing in the house, child's birth order, location in residence for cooking and type of cooking fuel used.

## Data analysis

The children's data set was imported into Stata 16 and merged with the household data set [25]. To produce nationally representative results, sample weights provided in the datasets were used. Children with incomplete interviews were excluded from the initial sample (S1 Fig in S1 File). Chi Squared descriptive univariable analyses were conducted for RARI and PCV uptake to explore significant associations between the dependent variables and the social economic independent variables. A chi Squared analysis is appropriate for categorical data to statistically analyse frequency distribution [28].

More detailed analysis was then conducted with uni and multiple logistic regression models to adjust for independent confounding variables, calculating odds ratios (ORs) with a 95% confidence interval (CI) for RARI associations. Logistic regression analysis associates the ratio of the odds of an event occurring, given the value of an independent variable compared to the reference category of the independent variable, e.g. the odds of RARI if rural living compared to urban living. Multiple logistic regression models examine the impact of multiple variables accounting for several potentially confounding variables simultaneously and is the appropriate regression analysis when the dependent variable is binary [29] as in this instance. The same analysis was also completed comparing variables against PCV 1/2/3 doses. A p value of 0.05 or less (2 sided) was considered significant for all statistical tests. A p value quantifies the significance of an association and the 95% CI quantifies the preciseness of the estimation with a values range [30] for which if the study was repeated multiple times, the true effect would be within this range 95% of the time [31].

## Results

### Survey population

28,479 households were selected for the sample, 27,030 of which were occupied and 26,713 successfully interviewed, a response rate of 99 percent [24]. 19,285 CU5 were listed as eligible for interview in the children's questionnaire, of which, 18,981 completed questionnaires and became the survey sample, a response rate of 98 percent (S1 Fig in S1 File).

Table 1 details the sample characteristics. The sample produced an almost equal male/female ratio with a mean average age of 29.9 months. 48.5% of the sample lived in the southern region, 34% central and 17.5% were from the northern region. 88.8% of the sample population was rural living. 13.3% of the children's mothers has no education, 70.2% had primary level education, 15.4% secondary and less than 1% had received higher education. 23% of children were categorised in the poorest and 22% in the second poorest wealth quintiles. The middle wealth quintile comprised of 21.5% of the sample, 18.6% in the fourth richest and 15.1% were in the richest quintile. Positively, 96.2% of children scored 'normal' in the WHO standard weight for height scores. 52.8% of children lived with less than 5 people, 44.8% lived with less than 10. Less than 2.1% lived with more than 10 people but 43 (0.2%) children resided with more than 16. 99.3% of children lived with less than 4 other children. Of those with data, 43.1% of the children surveyed were the first child to be born to their mother and 25.5% were

**Table 1. Sample characteristics.**

| Variable | Measure | Number of children Under 5 n = 18,981 |
|---|---|---|
| **Sex** | Male | n = 9,490 (50%) |
| | Female | n = 9,491 (50%) |
| | Missing | 0 |
| **Region** | Northern | n = 3,320 (17.5%) |
| | Central | n = 6,451 (34%) |
| | Southern | n = 9,210 (48.5%) |
| | Missing | 0 |
| **Area** | Urban | n = 2,125 (11.2%) |
| | Rural | n = 16,856 (88.8%) |
| | Missing | 0 |
| **Age** | Mean average age of sample 29.9 months | |
| | 0–5 months | n = 1,686 (8.8%) |
| | 6–11 months | n = 1,791 (9.4%) |
| | 12–23 months | n = 3,870 (20.3%) |
| | 24–35 months | n = 3,795 (19.9%) |
| | 36–47 months | n = 4,099 (21.6%) |
| | 48–59 months | n = 3,740 (19.7%) |
| | Missing | 0 |
| **Mothers education level** | None | n = 2,528 (13.3%) |
| | Primary | n = 13,330 (70.2%) |
| | Secondary | n = 2,921 (15.4%) |
| | Higher | n = 188 (1%) |
| | Missing | n = 14 (0.07%) |
| **Wealth quintile** | Poorest | n = 4,290 (23%) |
| | Second | n = 4,190 (22%) |
| | Middle | n = 4,082 (21.5%) |
| | Fourth | n = 3,538 (18.6%) |
| | Richest | n = 2,881 (15.1%) |
| | Missing | 0 |
| **Weight for height Z score** | >-2 Zscores (normal) | n = 18,269 (96.2%) |
| | -3 to -2 Zscores (low) | n = 508 (2.7%) |
| | <3 Zscores (severely low) | n = 204 (1.1%) |
| | Missing | 0 |
| **Number of people residing in household (from merged household data set)** | <5 | n = 10,023 (52.8%) |
| | 6–10 | n = 8,507 (44.8%) |
| | 11–25 | n = 451 (2.4%) |
| | Missing | 0 |
| **Number of children under 5 residing in house (from merged household data set)** | 1–3 | n = 18,840 (99.3%) |
| | 4–6 | n = 141 (0.7%) |
| | Missing | 0 |

(*Continued*)

**Table 1.** (Continued)

| Variable | Measure | Number of children Under 5 n = 18,981 |
|---|---|---|
| **Childs birth order (from merged household data set)** | 1st | n = 8,194 (43.2%) |
| | 2nd | n = 4,846 (25.5%) |
| | 3rd | n = 2,654 (14.0%) |
| | 4th | n = 1,570 (8.3%) |
| | 5th | n = 689 (3.6%) |
| | 6th | n = 298 (1.6%) |
| | 7th | n = 114 (0.6%) |
| | 8th | n = 34 (0.2%) |
| | Missing | n = 582 (3.1%) |
| **Location of cooking activities (from merged household data set)** | Separate kitchen room | n = 1,553 (8.1%) |
| | Elsewhere in house | n = 883 (4.6%) |
| | Separate building | n = 11,128 (58.6%) |
| | Outdoors | n = 5,240 (27.6%) |
| | Other | n = 18 (0.1%) |
| | Missing | n = 159 (0.8%) |
| **Cooking fuel used (from merged household data set)** | Electric | n = 153 (0.8%) |
| | Kerosene | n = 1 (0.01%) |
| | Coal | n = 3 (0.02%) |
| | Charcoal | n = 2,142 (11.3%) |
| | Wood | n = 16,611 (87.5%) |
| | Straw/shrubs | n = 46 (0.2%) |
| | Crops | n = 18 (0.1%) |
| | Other | n = 1 (0.01%) |
| | Missing | n = 6 (0.03%) |

the second child born. For nearly 59% of households, food was cooked in a separate kitchen or outside (27.6%). Of those with data, 87.5% of households used wood as cooking fuel (Table 1).

## RARI results

From the sample, 7,808 caregivers provided data on their children's RARI. 4,382 caregivers answered "yes" to their child RARI in the 2 weeks prior to the interview date, 3,426 responded "no", 11,173 results from the total CU5 sample were "missing" (S2 Fig in S1 File). Of the 11,173 missing data on RARI (variable CA8 "Difficulty breathing during illness with cough" in the MICS dataset, see questionnaire on page 637 of MICS [24]) only 4 responded "Don't Know" and 10 were marked as "Missing" with the other 11,159 missing data on this variable in the dataset. We therefore don't believe it's correct to assume that those without data are "No" (i.e. don't have RARI) and exclude them from our denominator. We report socio-economic characteristics of those with missing data on RARI separately in S1 Table in S1 File and find the proportion missing data on RARI only varies by 8 percentage points (e.g. for child age) or less across the categories of each socio-economic variable (S1 Table in S1 File), suggesting our analyses below are unlikely to be biased by this missing data. 56.2% of caregivers answering (4,382 / 7,808) RARI in their children in the 2 week interview recall period.

Of the 4,382 children answering yes to RARI, the 12–23 month old category reported the highest frequency of cases with 988 children affected. However, this age group was also the

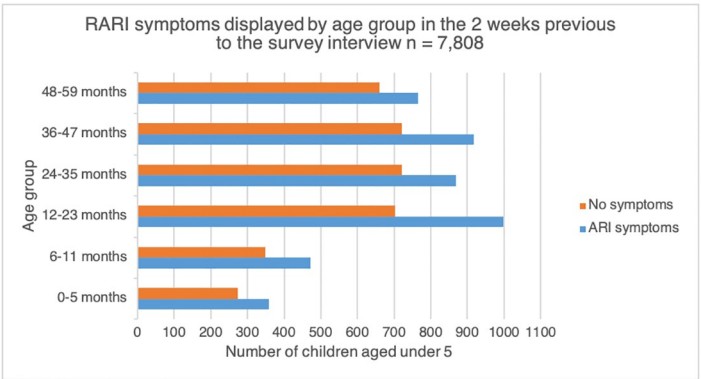

**Fig 1. Presenting RARI frequency by age group in months.**

largest in the sample (see Fig 1). Statistical analysis did not find the 6 month age group categories to be significantly associated with RARI (p = 0.074, Table 2).

All analytical models showed no significant relationship between sex and RARI. An independent association with RARI, adjusted multivariable logistic regression revealed children located in the central region had 18% reduced odds (OR 0.82, CI (0.72–0.93) p = 0.003) of RARI compared with the northern regions. Conversely, univariable analysis found children in the Southern region had increased odds of RARI (OR 1.18, CI (1.04–1.33) p = 0.008) compared to the northern region, however this association was not maintained when adjusted for other covariates. Rural dwelling children were also found to have significantly greater odds of RARI than urban dwellers in univariable logistic regression (OR 1.40, CI (1.21–1.62) p = 0.000) but not in multivariable models, alluding to rural dwelling not being independently associated with increased odds of RARI.

Chi squared analysis showed a significant relationship between RARI and a child's mothers education level (p = 0.006). Further exploration of this relationship with logistic regression analysis showed that only children of mothers who had received higher education were significantly less likely to RARI with 44% reduced odds (OR 0.57. CI (0.36–0.90) p = 0.016) compared to those of mothers with no education. However, when tested against other covariates no association between mother's education and RARI was found.

Mutlivariable logistic regression showed that children in the higher wealth quintiles had statistically significant increasingly reduced odds of RARI when compared to the poorest; middle (OR 0.86, CI (0.75–0.99) p = 0.039, fourth (OR 0.83, CI (0.71–0.96) p = 0.010, richest (OR 0.74, CI (0.61–0.89) p = 0.002). The WHO weight for height Z-score was not shown to be associated with RARI, this is not surprising as 96.2% of children scored within normal parameters. The number of people living in the house with was not associated with RARI, and neither was the number of children under 5 residing in the house, or child's birth order (Table 2).

There was no significant association between cooking location and RARI.

However, univariable analysis found a significant association between type of cooking fuel used and RARI (p = 0.000). Logistic regression found that those utilising the most frequently used fuel type, wood, to have 85% increased odds of RARI when compared to air pollution-free electric fuel (OR 1.85, CI (1.10–3.10) p = 0.019). This significance was not maintained when inputted into multivariable logistic regression suggesting confounding variables drove this result. Even after adjusting for covariates in multiple regression, children who lived in houses who burnt shrubs/straw as fuel had statistically significant much greater odds of RARI (OR 3.32, CI (1.07–10.28) p = 0.037) when compared to electric fuels. Other

**Table 2. Socio-economic associations with RARI in children under 5.**

| Exposure variable | Measure | Chi squared uni variable analysis: RARI | | Chi Squared Uni-analysis (95% CI) | Logistic regression analysis (95% CI) | Multi logistic regression analysis (95% CI)** |
|---|---|---|---|---|---|---|
| | | Yes | No | | | |
| **Sex** | **Male n = 3,931** | n = 2,218 (56.4%) | n = 1,713 (43.6%) | p = 0.589 | OR = 1 | - |
| | **Female n = 3,877** | n = 2,164 (55.8%) | n = 1,713 44.2%) | | OR 0.98 CI (0.89–1.07) p = 0.589 | |
| | **Total n = 7,808** | n = 4,382 (56.1%) | n = 3,426 (43.9%) | | | |
| **Region** | **Northern n = 1,486** | n = 825 (55.5%) | n = 661 (44.4%) | p = 0.000* | OR = 1 | OR = 1 |
| | **Central n = 2,614** | n = 1,350 (51.6%) | n = 1,264 (48.4%) | | OR 0.86 CI (0.75–0.97) p = 0.017* | OR 0.82 CI (0.72–0.93) p = 0.003* |
| | **Southern n = 3,708** | n = 2,207 (59.5%) | n = 1,501 (40.5%) | | OR 1.18 CI (1.04–1.33) p = 0.008* | OR 1.12 CI (0.99–1.28) p = 0.073 |
| | **Total n = 7,808** | n = 4,382 (56.1%) | n = 3,426 (43.9%) | | - | - |
| **Area** | **Urban n = 816** | n = 397 (48.6%) | n = 419 (51.3%) | p = 0.000* | OR = 1 | OR = 1 |
| | **Rural n = 6,992** | n = 3,985 (57.0%) | n = 3,007 (43.0%) | | OR 1.40 CI (1.21–1.62) p = 0.000* | OR 1.14 CI (0.95–1.37) p = 0.151 |
| | **Total n = 7,808** | n = 4,382 (56.1%) | n = 3,426 (43.9%) | | - | - |
| **Age** | **0–5 months n = 632** | n = 359 (56.8%) | n = 273 (43.2%) | p = 0.074 | OR = 1 | - |
| | **6–11 months n = 820** | n = 472 (57.6%) | n = 348 (42.4%) | | OR 1.03 CI (0.84–1.27) p = 0.772 | - |
| | **12–23 months n = 1,701** | n = 998 (58.7%) | n = 703 (41.3%) | | OR 1.08 CI (0.90–1.30) p = 0.416 | - |
| | **24–35 months n = 1,590** | n = 869 (54.6%) | n = 721 (45.3%) | | OR 0.92 CI (0.76–1.10) p = 0.358 | - |
| | **36–47 months n = 1,639** | n = 918 (56%) | n = 721 44.0%) | | OR 0.97 CI (0.80–1.17) p = 0.732 | - |
| | **48–59 months n = 1,426** | n = 766 (53.7%) | n = 660 (46.3%) | | OR 0.88 CI (0.73–1.07) p = 0.194 | - |
| | **Total n = 7,808** | n = 4,382 (56.1%) | n = 3,426 (43.9%) | | - | - |

(*Continued*)

**Table 2.** (Continued)

| Exposure variable | Measure | Chi squared uni variable analysis: RARI | | Chi Squared Uni-analysis (95% CI) | Logistic regression analysis (95% CI) | Multi logistic regression analysis (95% CI)** |
|---|---|---|---|---|---|---|
| | | Yes | No | | | |
| Mothers education level | None n = 861 | n = 488 (56.7%) | n = 373 (43.3%) | p = 0.006* | OR = 1 | OR = 1 |
| | Primary n = 5,624 | n = 3,204 (57.0%) | n = 2,420 (43.0%) | | OR 1.01 CI (0.88–1.17) p = 0.872 | OR 1.08 CI (0.93–1.25) p = 0.318 |
| | Secondary n = 1,232 | n = 653 (53.0%) | n = 579 (47.0%) | | OR 0.86 CI (0.72–1.03) p = 0.097 | OR 1.05 CI (0.87–1.26) p = 0.617 |
| | Higher n = 82 | n = 35 (42.7%) | n = 47 (57.3%) | | OR 0.57 CI (0.36–0.90) p = 0.016* | OR 0.87 CI (0.53–1.40) p = 0.549 |
| | Total n = 7,799 | n = 4,380 (56.2%) | n = 3,419 (43.8%) | | - | - |
| Wealth quintile | Poorest n = 1,726 | n = 1,027 (59.5%) | n = 699 (40.5%) | p = 0.000* | OR = 1 | OR = 1 |
| | Second n = 1,756 | n = 1,040 (59.2%) | n = 716 (40.8%) | | OR 0.99 (CI 0.86–1.13) p = 0.868 | OR 0.99 CI (0.86–1.13) p = 0.866 |
| | Middle n = 1,672 | n = 936 (56.0%) | n = 736 (44.0%) | | OR 0.87 CI (0.76–0.99) p = 0.038* | OR 0.86 CI (0.75–0.99) p = 0.039* |
| | Fourth n = 1,480 | n = 804 (54.3%) | n = 676 (45.7%) | | OR 0.81 CI (0.70–0.93) p = 0.003* | OR 0.83 CI (0.71–0.96) p = 0.010* |
| | Richest n = 1,174 | n = 575 (49.0%) | n = 599 (51.0%) | | OR 0.65 CI (0.56–0.76) p = 0.000* | OR 0.74 CI (0.61–0.89) p = 0.002* |
| | Total n = 7,808 | n = 4,382 (56.1%) | n = 3,426 (43.9%) | | - | - |
| WHO Weight for height Z score | >-2 Zscores (normal) n = 7,520 | n = 4,213 (56.0%) | n = 3,307 (44.0%) | p = 0.594 | OR = 1 | - |
| | -3 to -2 Zscores (low) n = 206 | n = 119 (57.8%) | n = 87 (42.2%) | | OR 1.07 CI (0.81–1.42) p = 0.619 | - |
| | <3 Zscores (severely low) n = 82 | n = 50 (61%) | n = 32 (39%) | | OR 1.23 CI (0.79–1.92) p = 0.370 | - |
| | Total n = 7,808 | n = 4,382 (56.1%) | n = 3,426 (43.9%) | | - | - |
| Number of people residing in house | <5 n = 4,204 | n = 2,364 (56.2%) | n = 1,840 (43.8%) | p = 0.836 | OR = 1 | - |
| | 6–10 n = 3,430 | n = 1,917 (55.9%) | n = 1,513 (44.1%) | | OR 0.99 CI (0.90–1.08) p = 0.764 | - |
| | 11–25 n = 174 | n = 101 (58.1%) | n = 73 (42.0%) | | OR 1.08 CI (0.79–1.46) p = 0.637 | - |
| | Total n = 7,808 | n = 4,382 (56.1%) | n = 3,426 (43.9%) | | - | - |

(*Continued*)

**Table 2.** (Continued)

| Exposure variable | Measure | Chi squared uni variable analysis: RARI | | Chi Squared Uni-analysis (95% CI) | Logistic regression analysis (95% CI) | Multi logistic regression analysis (95% CI)** |
|---|---|---|---|---|---|---|
| | | Yes | No | | | |
| Number of children under 5 residing in house | 1–3 n = 7,756 | n = 4,351 (56.1%) | n = 3,405 (43.9%) | p = 0.611 | OR = 1 | |
| | 4–6 n = 52 | n = 31 (59.6%) | n = 21 (40.4%) | | OR 1.16 CI (0.66–2.01) p = 0.611 | |
| | Totals n = 7,808 | n = 4,382 (56.1%) | n = 3,426 (43.9%) | | - | |
| Childs birth order | 1st n = 3,450 | n = 1,957 (56.7%) | n = 1,493 (43.3%) | p = 0.053 | treated as a linear variable. OR per unit increase: 1.00 CI (0.97, 1.04) p = 0.808 | |
| | 2nd n = 1,961 | n = 1,087 (55.4%) | n = 874 (44.6%) | | | |
| | 3rd n = 1,046 | n = 569 (54.4%) | n = 477 (45.6%) | | | |
| | 4th n = 636 | n = 348 (54.7%) | n = 288 (45.3%) | | | |
| | 5th n = 269 | n = 152 (56.5%) | n = 117 (43.5%) | | | |
| | 6th n = 123 | n = 85 (69.1%) | n = 38 (30.9%) | | | |
| | 7th n = 42 | n = 27 (64.3%) | n = 15 (35.7%) | | | |
| | 8th n = 13 | n = 5 (38.5%) | n = 8 (61.5%) | | | |
| | Totals n = 7,540 | n = 4,230 (56.1%) | n = 3,310 (43.9%) | | - | - |
| Location of cooking activities | Separate kitchen room n = 624 | n = 373 (59.8%) | n = 251 (40.2%) | p = 0.191 | OR = 1 | - |
| | Elsewhere in house n = 395 | n = 236 (59.7%) | n = 159 (40.3%) | | OR 1.00 CI (0.77–1.29) p = 0.993 | - |
| | Separate building n = 4,647 | n = 2,585 (55.6%) | n = 2,062 (44.4%) | | OR 0.84 CI (0.71–1.00) p = 0.050 | - |
| | Outdoors n = 2,072 | n = 1,157 (55.8%) | n = 915 (44.2%) | | OR 0.85 CI (0.71–1.02) p = 0.082 | - |
| | Other n = 8 | n = 4 (50%) | n = 4 (50%) | | OR 0.67 CI (0.17–2.71) p = 0.578 | - |
| | Totals n = 7,746 | n = 4,355 (56.2%) | n = 3,391 (43.8%) | | - | - |

*(Continued)*

**Table 2.** (Continued)

| Exposure variable | Measure | Chi squared uni variable analysis: RARI | | Chi Squared Uni-analysis (95% CI) | Logistic regression analysis (95% CI) | Multi logistic regression analysis (95% CI)** |
|---|---|---|---|---|---|---|
| | | Yes | No | | | |
| Cooking fuel used | Electric n = 60 | n = 25 (41.7%) | n = 35 (58.3%) | p = 0.000* | OR = 1 | OR = 1 |
| | Kerosene n = 1 | Not enough data for analysis n = 1 | | | Not enough data for analysis n = 1 | |
| | Coal n = 1 | Not enough data for analysis n = 1 | | | Not enough data for analysis n = 1 | |
| | Charcoal n = 857 | n = 426 (49.7%) | n = 431 (50.3%) | | OR 1.38 CI (0.81–2.35) p = 0.230 | OR 1.16 CI (0.67–2.00) p = 0.601 |
| | Wood n = 6,855 | n = 3,902 (56.9%) | n = 2,953 (43.1%) | | OR 1.85 CI (1.10–3.10) p = 0.019* | OR 1.25 CI (0.72–2.18) p = 0.429 |
| | Straw/shrubs n = 25 | n = 20 (80%) | n = 5 (20%) | | OR 5.60 CI (1.85–16.9) p = 0.002* | OR 3.32 CI (1.07–10.28) p = 0.037* |
| | Crops n = 8 | n = 6 (75%) | n = 2 (25%) | | OR 4.20 CI (0.78–22.55) p = 0.094 | OR 2.34 CI (0.43–12.75) p = 0.327 |
| | Other n = 1 | Not enough data for analysis n = 1 | | | Not enough data for analysis n = 1 | |
| | Totals n = 7,806 | n = 4,380 (56.1%) | n = 3,426 (43.9%) | | - | - |

*Indicates significance at the p <0.005 level

** Only the independent variables with a significant association at uni-variable analysis were included in the single multi logistic regression model.

fuels were not found to be significantly associated or lacked the sufficient sample size for analysis (Table 2).

## PCV results

Only 2% (n = 236) of the 11,462 children aged 1–36 months did not have data on whether or not they had a vaccination card therefore we do not report non-response separately in our PCV analysis. For children who did not have a vaccination card, their caregiver verbally reported if the child had ever received the PCV and, if answered yes, how many. Results showed 10.6% of children did not have a vaccination card, 10.8% had a card but it was not shown to the investigator and 78.6% presented their card for viewing. For those that did not have a card to evidence, 94.9% reported that their child had received at least one PCV vaccine whilst 5.1% had never had a PCV vaccine. 73.0% of children without a vaccine card reported having all 3 PCV, 17.1% had received 2 and 9.8% had only received 1 (S2 Table S1 File).

For children aged 1–36 months who had vaccination cards, PCV vaccination status and year given was recorded by the interviewer. All "yes" answers from each year, 2011, 2012, 2013, 2014 were combined to give an overall indication of the number of children who had received the PCV from their vaccination cards (S3 Table S1 File).

To summarise all the children aged between 1–36 months who had received PCV vaccinations into a numerical variable, the verbally reported responses and vaccination card responses were combined to produce the new PCVtotals variable for each of the three doses to enable analysis (S3 Fig in S1 File).

**Table 3. PCV coverage calculations and result estimates.**

| Total number of children aged 1–36 months surveyed about their PCV status | % Vaccine coverage for 1–36 month olds between November 2011—April 2014 |
|---|---|
| **PCV1** | |
| n = combining PCV1 total sample number from record card and verbal report: 8,449 + 2,049 = 10,498 | (100/10498) x 9797 = **93.3%** |
| **PCV2** | |
| n = combining PCV2 total sample number from record card and verbal report: 8,426 + 2,049 = 10,475 | (100/10475)x 9,096 = **86.8%** |
| **PCV3** | |
| n = combining PCV3 total sample number from record card and verbal report: 8,362 + 2,049 = 10,411 | (100/10411)x 8,016 = **77.0%** |

## PCV vaccine coverage

Using the PVCtotals variable, vaccine coverage estimates were calculated for each PCV dose for children aged 1–36 months. We estimate 93.3% of 1–36 month olds to have received PCV1, 86.8% to have received PCV2 and 77.0% to have received all 3 doses (Table 3).

Using the PCVtotals variable we were able to compare the average age a child received each PCV dose in our sample for the years 2011/12/13/14. Results show the average age for a child to receive their first dose of PCV has declined from 30.5 months in 2011 to 2.9 months in 2014, a reduction of 26.6 months. The average age to receive the 2nd PCV dose has reduced from 30.8 to 4.4 months old, a reduction of 26.4 months and the average age of the 3rd dose also declined by 26.1 months from 31.8 to 5.7 months old (See Fig 2).

No significant association was found in analysis between sex and any PCV dose (see Table 4 for all PCV1/2/3 results). When compared to the northern region, uni-variable analysis found southern dwelling children to have 16% reduced odds of receiving PCV1 (OR 0.84, CI (0.72–0.99) p = 0.035*) 22% reduced odds of PCV2 (OR 0.78, CI (0.68–0.89) p = 0.000*) and 20% reduced odds of receiving PCV3 (OR 0.80, CI (0.71–0.90) p = 0.000*). Only PCV2 maintained a statistically significant odds reduction association for southern dwellers when adjusted for confounders in multi-variable analysis (OR 0.85, CI (0.72–0.996) p = 0.044*). When comparing the central region to the north, logistic regression found children living in the central

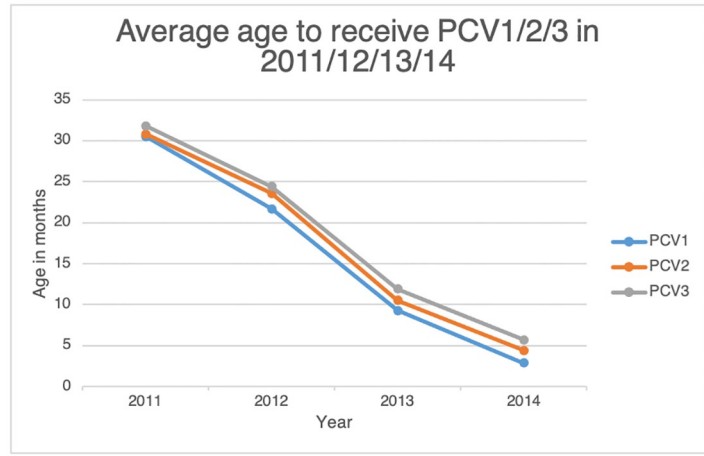

**Fig 2. Demonstrating a child's average age in months to receive PCV/1/2/3 since introduction in 2011 and survey completion in 2014.**

**Table 4. Socio-economic associations with PCV1/2/3 for 1–36 month olds.**

| Exposure variable | PCV1 Chi Squared uni-variable analysis Yes | PCV1 Chi Squared uni-variable analysis No | Chi Squared PCV1 (95% CI) | Logistic regression PCV1 (95% CI) | Multi variable logistic regression PVC1 (95% CI) | PCV2 Chi Squared uni-variable analysis Yes | PCV2 Chi Squared uni-variable analysis No | Chi Squared PCV2 (95% CI) | Logistic regression PCV2 (95% CI) | Multi variable logistic regression PVC2 (95% CI) | PCV3 Chi Squared uni-variable analysis Yes | PCV3 Chi Squared uni-variable analysis No | Chi Squared PVC3 (95% CI) | Logistic regression PCV3 (95% CI) | Multi variable logistic regression PVC3 (95% CI) |
|---|---|---|---|---|---|---|---|---|---|---|---|---|---|---|---|
| **Sex n = 11,287** | | | | | | | | | | | | | | | |
| Male n = 5,640 | n = 4,920 (87.2%) | n = 720 (12.8%) | p = 0.172 | OR = 1 | - | n = 4,572 (81.1%) | n = 1,068 (18.9%) | p = 0.202 | OR = 1 | - | n = 4,000 (70.9%) | n = 1,640 (29.1%) | p = 0.819 | OR = 1 | - |
| Female n = 5,647 | n = 4,877 (86.4%) | n = 770 (13.6%) | | OR 0.93 CI (0.83–1.03) p = 0.172 | | n = 4,524 (80.1%) | n = 1,123 (19.9%) | | OR 0.94 CI (0.86–1.03) p = 0.202 | | n = 4,016 (71.1%) | n = 1,631 (28.9%) | | OR 1.01 CI (0.93–1.10) p = 0.819 | |
| **Region n = 11,287** | | | | | | | | | | | | | | | |
| Northern n = 1,898 | n = 1,667 (87.8%) | n = 231 (12.2%) | p = 0.021* | OR = 1 | OR = 1 | n = 1,581 (83.3) | n = 317 (16.7%) | p = 0.001* | OR = 1 | OR = 1 | n = 1,412 (74.4%) | n = 486 (25.6%) | p = 0.001* | OR = 1 | OR = 1 |
| Central n = 3,835 | n = 3,359 (87.6%) | n = 476 (12.4%) | | OR 0.98 CI (0.83–1.16) p = 0.794 | OR 1.12 CI (0.93–1.35) p = 0.233 | n = 3,102 (80.9%) | n = 733 (19.1%) | | OR 0.85 CI (0.73–0.98) p = 0.026* | OR 0.96 CI (0.81–1.14) p = 0.645 | n = 2,725 (71.1%) | n = 1,110 (28.9%) | | OR 0.84 CI (0.75–0.96) p = 0.008* | OR 0.97 CI (0.83–1.12) p = 0.641 |
| Southern n = 5,554 | n = 4,771 (85.9%) | n = 783 (14.1%) | | OR 0.84 CI (0.72–0.99) p = 0.035* | OR 0.94 CI (0.79–1.12) p = 0.486 | n = 4,413 (79.5%) | n = 1,141 (20.5%) | | OR 0.78 CI (0.68–0.89) p = 0.000* | OR 0.85 CI (0.72–0.996) p = 0.044* | n = 3,879 (69.8%) | n = 1,675 (30.2%) | | OR 0.80 CI (0.71–0.90) p = 0.000* | OR 0.88 CI (0.76–1.01) p = 0.078 |
| **Area n = 11,287** | | | | | | | | | | | | | | | |
| Urban n = 1,263 | n = 1,140 (90.3%) | n = 123 (9.7%) | p = 0.000* | OR = 1 | OR = 1 | n = 1,065 (84.3%) | n = 198 (15.7%) | p = 0.000* | OR = 1 | OR = 1 | n = 942 (74.6%) | n = 321 (25.4%) | p = 0.003* | OR = 1 | OR = 1 |
| Rural n = 10,024 | n = 8,657 (86.4%) | n = 1,367 (13.6%) | | OR 0.68 CI (0.56–0.83) p = 0.000* | OR 0.76 CI (0.59–0.96) p = 0.024* | n = 8,031 (80.1%) | n = 1,993 (19.9%) | | OR 0.75 CI (0.64–0.88) p = 0.000* | OR 0.92 CI (0.75–1.13) p = 0.435 | n = 7,074 (70.6%) | n = 2,950 (29.4%) | | OR 0.82 CI (0.71–0.93) p = 0.003* | OR 1.04 CI (0.87–1.25) p = 0.653 |
| **Age group n = 11,287** | | | | | | | | | | | | | | | |
| 1–5 months n = 1,511 | n = 1,016 (67.2%) | n = 495 (32.8%) | p = 0.000* | OR = 1 | OR = 1 | n = 688 (44.9%) | n = 833 (55.1%) | p = 0.000* | OR = 1 | OR = 1 | n = 328 (21.7%) | n = 1,183 (78.3%) | p = 0.000* | OR = 1 | OR = 1 |
| 6–11 months n = 1,791 | n = 1,737 (96.9%) | n = 54 (3.0%) | | OR 15.7 CI (11.7–21.0) p = 0.000* | OR 15.9 CI (11.9–21.3) p = 0.000* | n = 1,682 (93.9%) | n = 109 (6.1%) | | OR 19.0 CI (15.2–23.6) p = 0.000* | OR 19.4 CI (15.6–24.1) p = 0.000* | n = 1,528 (85.3%) | n = 263 (14.7%) | | OR 21.0 CI (17.5–25.1) p = 0.000* | OR 21.8 CI (18.2–26.1) p = 0.000* |
| 12–23 months n = 3,870 | n = 3,701 (95.6%) | n = 169 (4.4%) | | OR 10.7 CI (8.8–12.9) p = 0.000* | OR 10.9 CI (9.0–13.1) p = 0.000* | n = 3,621 (93.6%) | n = 249 (6.4%) | | OR 17.9 CI (15.2–21.0) p = 0.000* | OR 18.3 CI (15.5–21.6) p = 0.000* | n = 3,406 (88.1%) | n = 464 (12.0%) | | OR 26.5 CI (22.6–30.9) p = 0.000* | OR 27.5 CI (23.5–32.2) p = 0.000* |
| 24–35 months n = 3,795 | n = 3,262 (86.0%) | n = 533 (14.0%) | | OR 3.0 CI (2.6–3.4) p = 0.000* | OR 3.03 CI (2.63–3.50) p = 0.000* | n = 3,047 (80.3%) | n = 748 (19.7%) | | OR 5.0 CI (4.4–5.7) p = 0.000* | OR 5.08 CI (4.46–5.79) p = 0.000* | n = 2,694 (71.0%) | n = 1,101 (29.0%) | | OR 8.83 CI (7.66–10.2) p = 0.000* | OR 9.09 CI (7.89–10.5) p = 0.000* |
| **Mothers education level n = 11,277** | | | | | | | | | | | | | | | |

*(Continued)*

Table 4.  (Continued)

| Exposure variable | PCV1 Chi Squared uni-variable analysis | | Chi Squared PCV1 (95% CI) | Logistic regression PCV1 (95% CI) | Multi variable logistic regression PVC1 (95% CI) | PCV2 Chi Squared uni-variable analysis | | Chi Squared PCV2 (95% CI) | Logistic regression PCV2 (95% CI) | Multi variable logistic regression PVC2 (95% CI) | PCV3 Chi Squared uni-variable analysis | | Chi Squared PCV3 (95% CI) | Logistic regression PCV3 (95% CI) | Multi variable logistic regression PVC3 (95% CI) |
|---|---|---|---|---|---|---|---|---|---|---|---|---|---|---|---|
| | Yes | No | | | | Yes | No | | | | Yes | No | | | |
| None n = 1,345 | n = 1,134 (84.3%) | n = 211 (15.7%) | p = 0.002* | OR = 1 | OR = 1 | n = 1,040 (77.3%) | n = 305 (22.7%) | p = 0.000* | OR = 1 | OR = 1 | n = 886 (65.9%) | n = 459 (34.1%) | p = 0.000* | OR = 1 | OR = 1 |
| Primary n = 7,966 | n = 6,910 (86.7%) | n = 1,056 (13.3%) | | OR 1.22 CI (1.04–1.43) p = 0.016* | OR 1.08 CI (0.90–1.29) p = 0.435 | n = 6,396 (80.3%) | n = 1,570 (19.7%) | | OR 1.19 CI (1.04–1.37) p = 0.012* | OR 1.06 CI (0.90–1.25) p = 0.465 | n = 5,619 (70.5%) | n = 2,347 (29.5%) | | OR 1.24 CI (1.10–1.40) p = 0.001* | OR 1.16 CI (1.01–1.34) p = 0.079 |
| Secondary n = 1,845 | n = 1,636 (88.7%) | n = 209 (11.3%) | | OR 1.46 CI (1.19–1.79) p = 0.000* | OR 1.16 CI (0.91–1.48) p = 0.240 | n = 1,542 (83.6%) | n = 303 (16.4%) | | OR 1.49 CI (1.25–1.78) p = 0.000* | OR 1.21 CI (0.98–1.51) p = 0.082 | n = 1,399 (75.8%) | n = 446 (24.2%) | | OR 1.63 CI (1.39–1.90) p = 0.000* | OR 1.52 CI (1.25–1.84) p = 0.000* |
| Higher n = 121 | n = 110 (90.9%) | n = 11 (9.1%) | | OR 1.86 CI (0.98–3.51) p = 0.056 | OR 0.996 CI (0.48–2.05) p = 0.990 | n = 110 (90.9%) | n = 11 (9.1%) | | OR 2.93 CI (1.56–5.52) p = 0.001* | OR 1.77 CI (0.86–3.62) p = 0.120 | n = 105 (86.8%) | n = 16 (13.2%) | | OR 3.40 CI (1.99–5.82) p = 0.000* | OR 2.68 CI (1.43–5.04) p = 0.002* |
| **Wealth quintile n = 11,287** | | | | | | | | | | | | | | | |
| Poorest n = 2,648 | n = 2,251 (85.0%) | n = 379 (15.0%) | p = 0.001* | OR = 1 | OR = 1 | n = 2,060 (77.8%) | n = 588 (22.2%) | p = 0.000* | OR = 1 | OR = 1 | n = 1,784 (67.4%) | n = 864 (32.6%) | p = 0.000* | OR = 1 | OR = 1 |
| Second n = 2,469 | n = 2,164 (86.7%) | n = 332 (13.3%) | | OR 1.15 CI (0.98–1.35) p = 0.082 | OR 1.14 CI (0.96–1.36) p = 0.141 | n = 1,996 (80.0%) | n = 500 (20.0%) | | OR 1.14 CI (0.996–1.30) p = 0.057 | OR 1.13 CI (0.97–1.32) p = 0.116 | n = 1,749 (70.1%) | n = 747 (29.9%) | | OR 1.13 CI (1.01–1.28) p = 0.037* | OR 1.13 CI (0.98–1.30) p = 0.083 |
| Middle n = 2,421 | n = 2,105 (87.0%) | n = 316 (13.1%) | | OR 1.17 CI (1.00–1.38) p = 0.047* | OR 1.24 CI (1.03–1.48) p = 0.020* | n = 1,944 (80.3%) | n = 477 (19.7%) | | OR 1.16 CI (1.02–1.33) p = 0.029* | OR 1.22 CI (1.04–1.43) p = 0.015* | n = 1,718 (71.0%) | n = 703 (29.0%) | | OR 1.18 CI (1.05–1.33) p = 0.006* | OR 1.24 CI (1.07–1.43) p = 0.003* |
| Fourth n = 2,042 | n = 1,774 (86.9%) | n = 268 (13.1%) | | OR 1.17 CI (1.00–1.38) p = 0.047* | OR 1.13 CI (0.93–1.36) p = 0.222 | n = 1,672 (81.9%) | n = 370 (18.1%) | | OR 1.29 CI (1.12–1.49) p = 0.001* | OR 1.26 CI (1.07–1.50) p = 0.007* | n = 1,488 (72.9%) | n = 554 (27.1%) | | OR 1.30 CI (1.15–1.48) p = 0.000* | OR 1.27 CI (1.09–1.48) p = 0.002* |
| Richest n = 1,680 | n = 1,503 (89.5%) | n = 177 (10.5%) | | OR 1.50 CI (1.24–1.81) p = 0.000* | OR 1.35 CI (1.05–1.72) p = 0.018* | n = 1,424 (84.8%) | n = 256 (15.2%) | | OR 1.59 CI (1.35–1.87) p = 0.000* | OR 1.57 CI (1.26–1.95) p = 0.000* | n = 1,277 (76.0%) | n = 403 (24.0%) | | OR 1.53 CI (1.34–1.76) p = 0.000* | OR 1.54 CI (1.27–1.88) p = 0.000* |
| **Number of people living in house n = 11,287** | | | | | | | | | | | | | | | |
| < 5 n = 6,215 | n = 5,419 (87.2%) | n = 796 (12.8%) | p = 0.133 | OR = 1 | - | n = 5,032 (81.0%) | n = 1,183 (19.0%) | p = 0.478 | OR = 1 | - | n = 4,457 (71.7%) | n = 1,758 (28.3%) | p = 0.158 | OR = 1 | - |
| 6–10 n = 4,810 | n = 4,144 (86.2) | n = 666 (13.9%) | | OR 0.91 CI (0.82–1.02) p = 0.111 | - | n = 3,857 (80.2%) | n = 953 (19.8%) | | OR 0.95 CI (0.87–1.05) p = 0.305 | - | n = 3,380 (70.3%) | n = 1,430 (29.7%) | | OR 0.93 CI (0.86–1.01) p = 0.097 | - |
| 11–25 n = 262 | n = 234 (89.3%) | n = 28 (10.7%) | | OR 1.23 CI (0.82–1.83) p = 0.314 | - | n = 207 (79.0%) | n = 55 (21.0%) | | OR 0.88 CI (0.65–1.20) p = 0.430 | - | n = 179 (68.3%) | n = 83 (31.7%) | | OR 0.85 CI (0.65–1.11) p = 0.233 | - |
| **Number of children U5 living in house n = 11,287** | | | | | | | | | | | | | | | |

(Continued)

**Table 4.** (Continued)

| Exposure variable | PCV1 Chi Squared uni-variable analysis | | Chi Squared PCV1 (95% CI) | Logistic regression PCV1 (95% CI) | Multi variable logistic regression PVC1 (95% CI) | PCV2 Chi Squared uni-variable analysis | | Chi Squared PCV2 (95% CI) | Logistic regression PCV2 (95% CI) | Multi variable logistic regression PVC2 (95% CI) | PCV3 Chi Squared uni-variable analysis | | Chi Squared PCV3 (95% CI) | Logistic regression PCV3 (95% CI) | Multi variable logistic regression PVC3 (95% CI) |
|---|---|---|---|---|---|---|---|---|---|---|---|---|---|---|---|
| | Yes | No | | | | Yes | No | | | | Yes | No | | | |
| **1–3 n = 11,210** | n = 9,734 (86.8%) | n = 1,476 (13.2%) | p = 0.195 | OR = 1 | | n = 9,044 (80.7%) | n = 2,166 (19.3%) | p = 0.004* | OR = 1 | OR = 1 | n = 7,971 (71.1%) | n = 3,239 (28.9%) | p = 0.015* | OR = 1 | OR = 1 |
| **4–6 n = 77** | n = 63 (81.8%) | n = 14 (18.2%) | | OR 0.68 CI (0.38–1.22) p = 0.198 | | n = 52 (67.5%) | n = 25 (32.5%) | | OR 0.50 CI (0.31–0.80) p = 0.004* | OR 0.48 CI (0.27–0.84) p = 0.010* | n = 45 (58.4%) | n = 32 (41.6%) | | OR 0.57 CI (0.36–0.90) p = 0.016* | OR 0.56 CI (0.33–0.96) p = 0.035* |

*Indicates significance at the p <0.005 level

** Only the independent variables with a significant association at uni variable analysis were included in the single multi logistic regression model

region to have reduced odds of receiving PCV2 (OR 0.85 CI (0.73–0.98) p = 0.026*) and PCV3 (OR 0.84, CI (0.75–0.96) p = 0.008*). This association was not significant in multivariable analysis and is therefore not independently associated with receiving PCV2.

For all 3 doses, uni-variable logistic models found rural children to have reduced odds of receiving the PCV compared to urban children PCV1 32% (OR 0.68, CI (0.56–0.83) p = 0.000*) PCV2 25% (OR 0.75, CI (0.64–0.88) p = 0.000*) PCV3 18% (OR 0.82, CI (0.71–0.93) p = 0.003*). Only the PCV1 association was statistically significant when adjusted for confounding variables in multi-variable analysis (OR: 0.76, CI (0.59–0.96) p = 0.024).

In all models, when compared to the 1–5 month age group, all age groups had statistically significant increased odds of PCV for all doses showing it to be independently associated with RARI, however the odds decreased with increasing age (see Table 4).

In uni-variable logistic regression for PCV1, mothers with primary education had a 22% increased odds of receiving PCV1 (OR 1.22, CI (1.04–1.43) p = 0.016*) and secondary education had a 46% increased odds of receiving PCV1 (OR 1.46, CI (1.19–1.79) p = 0.000*). In uni-variable logistic regression for PCV2, mothers with primary education had a 19% increased odds of receiving PCV2 (OR 1.19, CI (1.04–1.37) p = 0.012*), secondary education revealed a 49% increased odds of receiving PCV2 (OR 1.49, CI (1.25–1.78) p = 0.000*) which increased further with a higher education (OR 2.93 CI (1.56–5.52) p = 0.001*) when compared to no education. When adjusted for covariates, these associations between PCV1, PCV2 and a mother's education were not significant. Multi-variate logistic regression found children with mothers with a secondary (OR 1.52, CI (1.25–1.84) p = 0.000*) or higher education level (OR 2.68, CI (1.43–5.04) p = 0.002*) had statistically significant increased odds of receiving all 3 PCV doses when compared to no education.

For all 3 PCV doses, multi-variable logistic regression found children in the middle, fourth and richest wealth quintiles to have statistically significant increased odds of receiving the PCV vaccine when compared to the poorest (Table 4) making wealth independently associated with receiving all 3 PCV doses. The number of people residing in the house with the child was not found to have any statistically significant relationship with any PCV dose.

Mutli-variable logistic regression found children living with 4–6 CU5 to have decreased odds of receiving PCV2 and PCV3 when compared to children living in houses with 1–3 CU5 (PCV2 52% reduced odds (OR 0.48, CI (0.27–0.84) p = 0.010*) PCV3 44% reduced odds (OR 0.56, CI (0.33–0.96) p = 0.035*). This result makes CU5 living with 4–6 CU5 statistically significantly associated with reduced odds of receiving PCV doses (see Table 4 for all PCV1/2/3 results).

## Discussion

The findings of this study are of public health importance as it is the first we know of to associate nationally representative socio-economic factors with RARI in CU5 and PCV uptake in children aged 1–36 months in Malawi where the RARI burden is among the world's highest. Our study also estimated vaccine coverage in 2014 for 1–36 year olds and calculated the average age a child received PCV doses in 2011/12/13/14. Our study can inform programmers on where to provide targeted interventions to improve PCV uptake or avert ARI in the country.

The strengths of the study are its large sample size and that we used publicly available UNICEF MICS data, therefore, this study could be easily repeated by others when the next survey is complete, to reliably monitor progress.

### RARI

56% of children U5 in our community based sample RARI in the 2 week period prior to the survey interview. This is a very high percentage which is difficult to compare with others

studies that are region specific or who utilise different measures to identify ARI or clinically diagnose pneumonia in hospitals. One such example is a cluster randomised controlled trial by Mortimer et al [32] which reports an U5 pneumonia incidence rate of 15·67 per 100 child-years. A Malawi based study by Cox et al [6] estimated 2014's annual pneumonia prevalence for U5's at 32.6% (95% CI 29.3%- 36.0%), much below our estimations. These studies cannot be compared as Cox et al [6] collected data from the hospital health passports of 91.2% of their sample and true cases of pneumonia were diagnosed by health professionals, excluding cases of ARI, whereas in our study we categorised by ARI at house hold level.

Survey data has been found to produce over estimates which may explain why our retrospective 2 week prevalence percentage is so high. In this survey, RARI was identified on the basis of caregivers reporting 2 symptomatic indicators; a cough and fast or difficulty in breathing, which were not clearly defined or quantified, potentially causing high levels of measurement bias. Interviewers relied on the inherent limited ability of caregivers to correctly recognise and RARI. Prevalence estimates from this survey should be interpreted with caution as they are based on caregivers perception of symptoms and their capacity to recall events up to 2 weeks later, which may be prone to recall bias. Another consideration is that the ARI indicator does not record severity of symptoms or illness specificity, e.g. whether symptoms are due to asthma, infection or pneumonia or whether symptoms just need antibiotics or are life threatening. Acknowledging this, the MICS refers to ARI rather than pneumonia and this is why pneumonia estimates composed from survey data, such as the MICS, are thought to greatly inflate the number of pneumonia cases with most reported episodes being false positives [33].

Despite the flaws of RARI and assumed over estimation of pneumonia cases in survey data, the RARI indicator in MICS surveys is still deemed valid as it is a quick and simple measure applicable in the field [33]. Given the prevalence and high mortality rates from pneumonia, there is an urgent need for research to measure the sensitivity and specificity of RARI for the identification of true pneumonia cases. It has been proposed that it is possible to increase test specificity without negatively impacting the study design by adding a few additional symptoms or signs to survey questionnaires and employing a "pneumonia score" [34]. An evaluation of this measure is yet to be validated.

Our study found independent associations with reduced RARI to be central region living and children in the middle/fourth and richest wealth quintiles. Using straw/shrubs as fuel was found to be independently associated with increased RARI odds. Children living in the central region had reduced odds of RARI when compared to the north (OR 0.82, CI (0.71–0.93) p = 0.003). As there has been no nationally representative statistical analysis comparing regions associated with RARI or true pneumonia, we do not have any results to compare ours to. Our results suggest that further investigation is needed to examine reasons for regional discrepancies in RARI.

Our finding that using straw/shrubs as a cooking fuel increased the odds of RARI owing to poor air quality are not supported by a Malawian cluster RCT with an intervention of introducing a biomass stove by Mortimer et al [32]. The unexpected non-significant findings by Mortimer et al [32] were explained by exposure from other sources of air pollution such as rubbish burning and tobacco smoke, a potential confounder this study also could not account for. Despite this, our findings present evidence to deter families from burning straw/shrubs as it increases their odds of RARI.

Although not independently associated with RARI, that rural dwellers had increased odds of RARI in univariable analysis was not surprising when accounting for 95% of the poorest in Malawian living rurally [35] and our adjusted results demonstrating decreased odds of RARI for those in the highest wealth quintiles compared to the lowest. Conditions of poverty such as poor sanitation, lack of clean water and irregular hand washing cause a greater risk of pneumonia [36].

We found no association between RARI and sex, in difference with Cox et al [6] who reported that males are more likely to be affected. However, in Cox et al [6] cross-sectional study the male sample size was much larger than the female, the RARI sample cases small (n = 69) and the data was not weighted in analysis, which may explain this unexpected result.

No age group was found to have greater odds of RARI, however other studies have found higher ARI rates in children aged 6–23 months [7]. This difference is hard to compare as the study designs differed greatly. McCollum et al [7] conducted active pneumonia surveillance and actual pneumonia cases were diagnosed by clinicians, thus likely producing far more precise results than the generalised MICS data.

Our isolated, statistically significant, univariable logistic regression finding that children whose mother's attained a higher education had reduced odds of RARI is likely due to chance. A statistically significant p value of $p < 0.05$ still allows for 1 in 20 effects to be due to chance [31].

That cooking location was not significantly associated with RARI was unsurprising considering most of the sample cooked in a separate kitchen or outdoors, thus reducing air pollution which has been evidenced to cause pneumonia [37]. A major limitation of the survey is that it doesn't account for significant confounding comorbidities that predispose some children to pneumonia more than others, such as HIV [38]. Additionally, household surveys only include families with a permanent residence and can exclude those most vulnerable with less access to health care such as the homeless, displaced, children living in orphanages and nomadic families [39], thus possibly introducing selection bias.

## PCV

Our study found age group and wealth to be independent associated with 1–36 month olds receiving all 3 PCV doses. Secondary and higher educational attainment of mothers was independently associated with receiving 3 PCV doses only. Children living with 4–6 other children were found to independently have reduced odds of receiving PCV doses 2 and 3.

The advantages of using data from surveys such as the MICS is that that they are nationally representative, have large sample sizes, are of no cost to use and the instant obtainment of data via a download is very time effective. In LIC such as Malawi that lack accurate records, vaccine data often comes from caregiver interviews which risks recall and social biases. A strength of MICS surveys is that they collate information from vaccination cards and verbal recall when vaccination cards are not available so that there are no gaps in the data.

Positively, our results show the average age for a child to receive PCV1/2/3 has reduced by an average of 26.6 / 26.4 and 26.1 months, likely due to effective vaccine catch up campaigns. This study estimated 93.3% of 1–36 month olds to have received PCV1, 86.8% to have received PCV2 and 77.0% to have received all 3 doses. These findings are similar to the MICS [24] report of the same data set, but with some differences due to the MICS report separating estimates into age categories whereas this study averages 1–36 month olds. Another reason for the slight difference is that this studies estimates included recall data, whereas the MICS [24] report assumed that "For children without vaccination cards, the proportion of vaccinations given before the first birthday is assumed to be the same as for children with vaccination cards".

This study found 21% of children in our sample did not have a vaccination card or did not show it to the interviewer, however nearly 95% of these caregivers reported their children as having had at least one PCV. 21% is a large proportion of the sample, and therefore the opportunity for bias to reduce data accuracy is significant.

Although not a Malawian study or for the PCV, in Guatemala Goldman and Pebley [40] found that DTP coverage rates were similar for those with vaccine cards (70%) and those with cards but who were unable to present them (66.9%) whilst coverage estimates based on data

acquired from recall was much less at 48.5%. Although actual causation cannot be provided, these results were suggested to be accurate as they probably reflect reality, in that the absence of a vaccine card may be an indicator of infrequent contact with health care and therefore explains the lower vaccine coverage.

With no valid method for handling incomplete data and ensuring reliability of recall data, a systematic review of caregiver recall as a measure of vaccine coverage by Modi et al [41] concluded that recall data should be included to exclude bias and called for further research on increasing recall quality. A strength of our study is that it included recall data in PCV coverage estimates, in keeping with advice from existing research, potentially making our estimates more generalisable to the Malawian population and more accurate.

Although great improvements have been made, our results and other existing evidence shows that by April 2014, PCV coverage did not meet the Malawi Ministry of Health three-dose coverage target of 90% before 12 months. Our nationally representative study of children aged 1–36 months estimated that 77.0% of children had received all 3 PCV doses, less than the 86.0% estimated by Bondo et al [13], in rural children only, in one area (Kabadula) of one district (Lilongwe) of Malawi, aged between 6 weeks and 16 months. This could be explained by this studies findings in univariable logistic regression analysis that rural children have 0.82 reduced odds of receiving all 3 PCV vaccinations when compared to urban (p = 0.003, CI 0.71–0.93, however in multivariable regression there was no difference found, perhaps because of confounding of rural dwelling with other explanatory variables e.g. lower education, lower wealth quintile. That rural children were found to have decreased odds of PCV vaccination in univariable analysis is unsurprising and has previously been explained by longer distances to travel for health services [16].

The impact of our greatly reduced average age of vaccination between 2011 and 2014 is significant, as delayed vaccination extends susceptibility to ARI and reduces herd immunity [42]. Another positive confirmation from our results is that no association between sex and vaccination status was found, demonstrating that in Malawi one sex does not receive more favourable access to vaccinations.

Additionally, in multi-variable logistic analysis region was also found not to be associated with PCV vaccination, demonstrating the access to PCV vaccines is comparable across regions. An earlier study by Abebe et al [20] disagrees with this finding and found the northern region of Malawi to have lower vaccination rates. However, this study was based on data from 2007, before the PCV vaccine was introduced and optimistically, vaccine services may have been scaled up since then to produce our result. Regional differences often exist due to inequalities in health worker density and service provision [43]. Our finding of no significant regional difference is important as regional inequality of vaccine coverage can harbour clusters of under-vaccinated children, leading to an increased vulnerability of vaccine-preventable diseases such as pneumonia [20].

Given the vaccination catch up campaigns for those under 12 months and the advised vaccine delivery doses at 6,10 and 14 weeks for PCV/2/3 respectively, it is unsurprising that children older than 6 months were found to have increased odds of PCV uptake for all doses in all statistical tests with the highest uptake in the 6–11 month age category. Mvula et al [17], had similar findings as did McCollum et al [7] who found children aged 6–23 months old had the highest proportion of all 3 doses (71.7%, n = 8,567/11,948). This result is a promising find as it suggests greater vaccination coverage for children under 5 in the future and targets protecting children aged under 2 years old who are disproportionally affected by ARI with 80% of deaths attributed to ARI in this age group [44].

Once adjusted for confounding co-variables, only mothers with secondary or higher education had increased odds of a child receiving all 3 PCV does when compared to those with no

education. This finding emphasises the importance of maternal education on a child's health. It is thought educated women have great autonomy and control over household resources enhancing care seeking behaviours and their ability to comprehend health needs [45].

Across all doses, wealth greater than the 2 poorest quintiles was found to be associated with greater PCV uptake. This result is supported in the literature by Austin et al [18], and Zere et al [19], who reasoned that the poorest in society have less access to public health services with more barriers to health care such as accessing transport than wealthier parents.

Children living in houses with 4–6 children under 5 were found to have reduced odds of receiving PCV2 and PCV3 vaccine doses compared to those living in houses with 1–3 children under 5. A practical explanation for this could be that with so many other young children to care for, time resources are restricted which results in de-prioritising taking your child to be vaccinated. Dahiru [31] suggested that quantitative evidence cannot give us all the answers, and therefore we should always seek non-statistical evidence such as frequency distributions, theory and qualitative evidence wherever available to compare findings and fill the gaps. We suggest that qualitative research is needed to understand the reasons why some children are not vaccinated. Future research could also compare the next MICS survey with the 2015 one used in this study to look for trends. Additionally, the MICS and DHS data could be combined to increase sample size and generalisability, though care would be needed to ensure survey questions and methods are sufficiently equivalent.

## Conclusion

To achieve SDG targets and reduce preventable deaths of children under 5, it is essential Malawi addresses the high RARI burden. Whilst acknowledging the limitations of using survey data and the potential for recall bias, our cross-sectional analysis of secondary data reported social economic associations with RARI in children under 5 and PCV vaccine uptake for children aged 1–36 months. Our study estimated PCV vaccine coverage estimates for 2014 that were found to be below the >90% target but found significant decreases in average age a child received all 3 PCV vaccine doses since its introduction in 2011. Our findings highlight the importance of a mother's education on her child's health, the association between poverty and child health and recommends that straw/shrubs not be used as fuel.

Future studies are needed to validate a survey pneumonia specificity and sensitivity test that differentiates it from RARI. Qualitative research would aid understanding of the reasons children aren't vaccinated.

## Supporting information

**S1 File.**
(DOCX)

**S1 Data.**
(DO)

**S1 Dataset.**
(DTA)

## Author Contributions

**Conceptualization:** Justine Gosling, Tim Colbourn.

**Formal analysis:** Justine Gosling, Tim Colbourn.

**Investigation:** Justine Gosling, Tim Colbourn.

**Methodology:** Justine Gosling, Tim Colbourn.

**Project administration:** Justine Gosling.

**Resources:** Justine Gosling.

**Supervision:** Tim Colbourn.

**Validation:** Justine Gosling, Tim Colbourn.

**Visualization:** Justine Gosling.

**Writing – original draft:** Justine Gosling.

**Writing – review & editing:** Justine Gosling, Tim Colbourn.

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
