## [Decision Letter · Decision Letter 0]

7 Jun 2022

PONE-D-21-21665What predicts reported acute respiratory infection in children under 5 and PCV vaccination in children aged 1-36 months in Malawi? A secondary data analysis using the Malawi 2014 MICS surveyPLOS ONE

Dear Dr. Colbourn,

Thank you for submitting your manuscript to PLOS ONE. After careful consideration, we feel that it has merit but does not fully meet PLOS ONE’s publication criteria as it currently stands. Therefore, we invite you to submit a revised version of the manuscript that addresses the points raised during the review process.

We look forward to receiving your revised manuscript.

Kind regards,

Shinya Tsuzuki, MD, MSc

Academic Editor

PLOS ONE

Journal Requirements:

2. Please provide further information in the data availability statement regarding the source of the 2014 UNICEF Malawi MICS data (for example, where the data were obtained and if any permission was required to use) as per our data availability policy (https://journals.plos.org/plosone/s/data-availability#loc-acceptable-data-access-restrictions).

Additional Editor Comments (if provided):

Both reviewers made some useful comments then please respond each of them appropriately.

Reviewers' comments:

Reviewer's Responses to Questions

**Comments to the Author**

1. Is the manuscript technically sound, and do the data support the conclusions?

Reviewer #1: No

Reviewer #2: Yes

2. Has the statistical analysis been performed appropriately and rigorously? 

Reviewer #1: Yes

Reviewer #2: Yes

3. Have the authors made all data underlying the findings in their manuscript fully available?

Reviewer #1: Yes

Reviewer #2: Yes

4. Is the manuscript presented in an intelligible fashion and written in standard English?

Reviewer #1: No

Reviewer #2: Yes

5. Review Comments to the Author

Reviewer #1: Major comments

1. The manuscript seems to contain more than two separate research questions, or not conceptually connected to each other. As shown in the Title - “What predicts reported acute respiratory infection in children under 5” and PCV vaccination in children aged 1-36 months in Malawi, though the data source is the same, have separate objective, and also separately analyzed and discussed. Thus, the manuscript has too much information and difficult to follow/understand

2. Authors indicated nearly 60% non-response on RARI (In the result section). With such a huge non-response rate, the validity of the research finding, and hence the conclusion of the RARI and also the national representativeness of the research is highly questioned.

General comments

It is difficult to comment the manuscript at its current stage, as the manuscript contains chunk of information. But, generally the authors need to consider

1. The introduction section contains too much information which is not relevant for the paper/ lacks focusing on the objective. In addition, it has issues related with consistency/flow of information. The problem is not clear. Argument starts around “full vaccination” and “PCV’ going down the line.

3. Objective: Not usual to have four objectives (two primary and two secondary/specific) in scientific research papers. Hypotheses/ null hypothesis not clearly linked with the objectives

4. Methods: Too much explanation on the method used by UNICEF (MICS). Only the analysis section is inherent to this paper.

5. Results: The finding that, Over 56% of caregivers RARI in their children in the 2-week interview recall period, is very exaggerated because only the caregivers who respond to the question were included in the denominator.

The discussion and conclusion need to be reviewed based on the above, if the authors accept the comments.

Reviewer #2: The manuscript is of public health importance and was well written. However, major revisions have been suggested as regards study methods, analytic approach, results presentation and interpretations for consideration.

6. PLOS authors have the option to publish the peer review history of their article (what does this mean?). If published, this will include your full peer review and any attached files.

Reviewer #1: No

Reviewer #2: **Yes: **Ezekiel Mupere MBChB, MMed, MS., PhD

---

## [Author Response · Author response to Decision Letter 0]

4 Sep 2022

Please see our uploaded response to reviewers document.

---

## [Decision Letter · Decision Letter 1]

10 Nov 2022

PONE-D-21-21665R1What predicts reported acute respiratory infection in children under 5 and PCV vaccination in children aged 1-36 months in Malawi? A secondary data analysis using the Malawi 2014 MICS surveyPLOS ONE

Dear Dr. Colbourn,

Thank you for submitting your manuscript to PLOS ONE. After careful consideration, we feel that it has merit but does not fully meet PLOS ONE’s publication criteria as it currently stands. Therefore, we invite you to submit a revised version of the manuscript that addresses the points raised during the review process.

We look forward to receiving your revised manuscript.

Kind regards,

Shinya Tsuzuki, MD, MSc

Academic Editor

PLOS ONE

Additional Editor Comments:

Two reviewers raised further concerns and I agree with their opinion, then therefore I believe another round for revision would be appropriate for the manuscript.

Reviewers' comments:

Reviewer's Responses to Questions

**Comments to the Author**

1. If the authors have adequately addressed your comments raised in a previous round of review and you feel that this manuscript is now acceptable for publication, you may indicate that here to bypass the “Comments to the Author” section, enter your conflict of interest statement in the “Confidential to Editor” section, and submit your "Accept" recommendation.

Reviewer #1: (No Response)

Reviewer #3: (No Response)

2. Is the manuscript technically sound, and do the data support the conclusions?

Reviewer #1: Partly

Reviewer #3: Partly

3. Has the statistical analysis been performed appropriately and rigorously? 

Reviewer #1: No

Reviewer #3: No

4. Have the authors made all data underlying the findings in their manuscript fully available?

Reviewer #1: Yes

Reviewer #3: Yes

5. Is the manuscript presented in an intelligible fashion and written in standard English?

Reviewer #1: No

Reviewer #3: Yes

6. Review Comments to the Author

Reviewer #1: Dear authors/ editors

Thank you again for the opportunity to review the manuscript entitled "What predicts reported acute respiratory infection in children under 5 and PCV vaccination in children aged 1-36 months in Malawi? A secondary data analysis using the Malawi 2014 MICS survey"

I appreciate the authors for the improvements they made in the manuscript, particularly in the introduction section. Though, some editorial revisions are still needed. For instance,

• Fig 3 is presented while Fig 1 and 2 are not available

• Zero missing values in table 1 are not important,

• Ethical clearance is included under the data collection section, should be separated,

• The selection of independent variables was referenced to (see introduction), but the introduction didn’t adequately address it.

• Univariate analysis table shall be presented and discussed separately before the multivariate Table should

• The numbers for the tables and figures are not correct and shall be edited

Major comments

I would like to re-emphasize on the previous comments, based on the authors response and in line with the developments in the manuscript

1. For the comment of having four objectives (two primary and two secondary/specific) and Hypotheses/ null hypothesis not clearly linked with the objectives, the authors respond that they do not see a scientific problem here and would like to keep all 4 objectives, and the hypotheses are clearly linked to our two primary objectives.

I agree that no problem of having more than one research question in a scientific paper. My intention is, for you to consider including very specific objectives under one or two comprehensive objectives, as the objectives are very similar. This would help to focus your discussion and presentation and also ease for readers.

Otherwise, authors shall consider addressing the statement of the problem and methods for each objective; and present and discuss each objective separately.

2.Methods section: Too much explanation on the method used by UNICEF (MICS). Only the analysis section is inherent to this paper.

Authors response: We believe the explanation of the methods used by UNICEF (MICS) that we have briefly summarized in this paper are relevant to readers understanding of the data we used and would like to keep this information in the paper.

Comment: I agree, but while a brief summary of the methods used by MICS is important, the methods used to for this secondary data study is more important for readers. It is important to inform readers, the design, how and what data has been extracted and organized, the operational definition, inclusion and exclusion criteria, etc., for the four objectives.

For instance the study population for RARI are U5C and for PCV coverage 1-36 months children are not the same and so is for data analysis etc., Thus, the authors shall address the methods (all subsections) used for each objective, as well as how the predictors of RARI and PCV uptake are selected

Results: The finding that, Over 56% of caregivers RARI in their children in the 2-week interview recall period, is very exaggerated because only the caregivers who respond to the question was included in the denominator.

Authors response: Thanks for highlighting this. We recognize that caregiver response to the

question on RARI in the previous two-weeks may be subject to some selection bias and this is already included as a limitation of our paper, as explained in the second paragraph of the RARI sub-section in our Discussion:

The previous comment is that the overestimation is created because the authors removed the participants who did not respond from the denominator, not due to selection bias or limitations inherent to survey methods.

The MICS 2014 reports 7.8% (~1475 of the 18981 children) with RARI (cough and fast/difficult breathing) were reported and that there was no non-response. Thus, how this study faced such a huge non-response rate, while using the same data? shall be explained and justified.

In addition, in such huge nonresponse rate, it is important to analyze the non-responses against the independent variables and explain to readers the characteristics of those who are missing and how it may affect the result.

PCV: Similar to RARI, the number/percent of children 1-36 months enrolled in PCV up take, and non-response rate should be also presented before going to the analysis.

PCV 3 coverage: The nationally representative study of children aged 1-36 months estimated that 77.0% of children had received all 3 PCV doses, less than the 86.0% estimated by Bondo et al., (2018) (PP 50). ?? Are the less than 3-month children expected to complete PCV 3 and included in the denominator? -

Reviewer #3: The manuscript by Gosling et al. reports the results of a study conducted to identify the predictors of RARI in children under 5 and the predictors of PCV in children aged 1-36 months in Malawi.

The topic of the study is important and interesting, however, the objective of the study, the statistical models and the outcome definitions are poorly described and require additional elaboration.

In general, the topic of the study is important and interesting, however, the manuscript is difficult to read and there are some methodological issues to deal with. All the following comments do not exclude that the results of the analysis are correct and interesting, but represent suggestions to improve the paper and mostly his presentation.

In particular:

• Abstract: Authors should report the objectives of the study in the abstract.

• Objectives:

o The definition of the study aims need to be revised. In particular, authors reported: “1.The national, social economic predictors for RARI in children under 5; 2. The national, social economic predictors of PCV uptake in children aged 1-36 months. The 1-36 month age range reflects the available data from the MICS survey.” I suggest to delete this last sentence (The 1-36 month age range reflects the available data from the MICS survey.) and to put it in the methods paragraph.

o For what concern the secondary objectives, they are not really objectives but results that could be reported in the results paragraph as part of the sample description.

• Methods:

o Could authors explain why they decided to consider children age as a predictor of PCV? I think that is difficult to interpret these results also because the dose of PCV (PCV1, PCV2, PCV3) depends on child age.

o About the analysis of the data, I suggest to consider the possibility of putting together some variables categories that have not enough numerosity. For example, for mother education, authors could consider secondary and higher together; for number of people residing in house they can consider 11+ as a category; child’s birth order could be considered as linear and cooking fuel used could be recategorized. I this way the results would be more clears and stable.

o In the data analysis paragraph, authors reported explanation about the chi squared test, the logistic regression and the p-value: “A chi Squared analysis is appropriate for categorical data to statistically analyse frequency distribution (Sharpe 2015).”; “Logistic regression analysis predicts the ratio of the odds of an event occurring, given the value of an independent variable compared to the reference category of the independent variable, e.g. the odds of RARI if rural living compared to urban living. Multiple logistic regression models examine the impact of multiple variables accounting for several potentially confounding variables simultaneously and is the appropriate regression analysis when the dependent variable is binary (Christiansen et al, 2015) as in this instance.” “A p value quantifies the significance of an association and the 95% CI quantifies the preciseness of the estimation with a values range (Kim and Bang, 2016) for which if the study was repeated multiple times, the true effect would be within this range 95% of the time (Dahiru 2008).” I think that is not necessary to explain them in a scientific paper, what they should reported is how they use these methods, which are the dependent and independent variables, as they done in the previous paragraphs.

o In the same way I suggest to delete the paragraphs related to hypotheses and null hypotheses, they are unnecessary if you have well described the objectives of your paper.

• Results:

o Tables 2 and 6 are very difficult to read; if authors believe that is important to show all results and all steps of the analysis, I suggest to report a table with chi squared test with all variables and a table with the complete model, reporting crude and udjusted OR of the logistic regression analysis. Otherwise, I suggest to put the information about all tested variables that resulted not associated, only in the text or in additional material.

o In the results paragraph (at page 36), authors reported: “In all models, when compared to the 1-5 month age group, all age groups had statistically significant increased odds of PCV for all doses showing it to be an independent predictor of RARI, however the odds decreased with increasing age”. Is not expected that the risk of being vaccinated increase with increasing age? Why authors do not analyze the risk of being vaccinated to PCV regardless to the number of doses?

o I have some concern about the region (northern, central and southern) and the area (urban or rural); There are urban and rural areas in all regions or these variables are showing the same things?

o In this paper authors took into account several predictors of RARI and PCV, some of these seem to represent similar aspects. Did authors evaluate collinearity or effect modification before establishing if these variables could be or not predictors of the outcomes?

MINOR COMMENTS:

• Tables titles need to be revised; for example, title of table 1 could be “Sample characteristics”; for table 2 (but also for others similar tables) I suggest to report something like “Socio economic predictors for RARI in children under 5 - results from the logistic regression model”.

• I do not understand as authors numbered figures in the paper. Figures 3 is the first figure of the paper, table 5 is after table 2.

7. PLOS authors have the option to publish the peer review history of their article (what does this mean?). If published, this will include your full peer review and any attached files.

Reviewer #1: **Yes: **Atakelti A Derbew

Reviewer #3: No

---

## [Author Response · Author response to Decision Letter 1]

23 Dec 2022

Please see our response to reviewers document and revised paper

---

## [Decision Letter · Decision Letter 2]

30 Jan 2023

PONE-D-21-21665R2What predicts reported acute respiratory infection in children under 5 and PCV vaccination in children aged 1-36 months in Malawi? A secondary data analysis using the Malawi 2014 MICS surveyPLOS ONE

Dear Dr. Colbourn,

Thank you for submitting your manuscript to PLOS ONE. After careful consideration, we feel that it has merit but does not fully meet PLOS ONE’s publication criteria as it currently stands. Therefore, we invite you to submit a revised version of the manuscript that addresses the points raised during the review process.

Please submit your revised manuscript by Mar 16 2023 11:59PM If you will need more time than this to complete your revisions, please reply to this message or contact the journal office at plosone@plos.org. Please include the following items when submitting your revised manuscript:A rebuttal letter that responds to each point raised by the academic editor and reviewer(s). You should upload this letter as a separate file labeled 'Response to Reviewers'.A marked-up copy of your manuscript that highlights changes made to the original version. You should upload this as a separate file labeled 'Revised Manuscript with Track Changes'.An unmarked version of your revised paper without tracked changes. You should upload this as a separate file labeled 'Manuscript'.If applicable, we recommend that you deposit your laboratory protocols in protocols.io to enhance the reproducibility of your results. Protocols.io assigns your protocol its own identifier (DOI) so that it can be cited independently in the future. For instructions see: https://journals.plos.org/plosone/s/submission-guidelines#loc-laboratory-protocols. Additionally, PLOS ONE offers an option for publishing peer-reviewed Lab Protocol articles, which describe protocols hosted on protocols.io. Read more information on sharing protocols at https://plos.org/protocols?utm_medium=editorial-email&utm_source=authorletters&utm_campaign=protocols.

We look forward to receiving your revised manuscript.

Kind regards,

Shinya Tsuzuki, MD, MSc

Academic Editor

PLOS ONE

Journal Requirements:

Additional Editor Comments:

Unfortunately original reviewers had already faded away, but new reviewers added several comments based on the original review. I think their suggestions are also reasonable, then minor revision will be required.

Reviewers' comments:

Reviewer's Responses to Questions

**Comments to the Author**

1. If the authors have adequately addressed your comments raised in a previous round of review and you feel that this manuscript is now acceptable for publication, you may indicate that here to bypass the “Comments to the Author” section, enter your conflict of interest statement in the “Confidential to Editor” section, and submit your "Accept" recommendation.

Reviewer #4: All comments have been addressed

Reviewer #5: (No Response)

2. Is the manuscript technically sound, and do the data support the conclusions?

Reviewer #4: Yes

Reviewer #5: Partly

3. Has the statistical analysis been performed appropriately and rigorously? 

Reviewer #4: Yes

Reviewer #5: Yes

4. Have the authors made all data underlying the findings in their manuscript fully available?

Reviewer #4: Yes

Reviewer #5: No

5. Is the manuscript presented in an intelligible fashion and written in standard English?

Reviewer #4: No

Reviewer #5: Yes

6. Review Comments to the Author

Reviewer #4: I can see that the manuscript have already gone through a multiple round of review. I still have few suggestions and doubts which as follows:

What is MICS data. Author need to give full form and explain the type of mics survey and data.Why they did not use complete DHS survey?

The ethical statement should be that Author have used secondary data available in public domain and does not require ethical approval. Currently ethical statement is confusing and funding statement should also be separate.

It seems like the sections can be made little concise for example: Author does not need to state every stats from the table, they should provide important details which is important and they want to highlight. Stating wealth index or some region is x percentage does not add value it make it more verbose.

As a reader, Some one would get lost in so many obvious percentages reported for example: the wealth index should be around 20 percent because the way it gets divided into quintiles. What are the regions and why it is important?

Fig 1: the Percentages in each group with symptoms will be more useful rather than providing numbers

Why the need of chi-square test when author are performing the logistic regression. Are they used it for selecting variable.

The results section is too verbose, it need to be more focused and provide answer to the objective and research question raised in the introduction

Limitation and strength will come at the end of discussion.

Conclusion: it is not surprising to found education and wealth to be related with child health. What more the analysis add in the context of malawi or what factors brought changes in children health in Malawi? will be more useful to conclude rather than giving a generic conclusion

Reviewer #5: This paper used 2014 UNICEF Malawi Multiple Cluster Indicator Survey to analyse socio economic predictors for RARI un CU% and PCV uptake in children aged 1-36 months. I hope the authors find the following comments and suggestions helpful.

Major comments:

1. I would advise that the authors use the term "association" instead of "predictor" in the title and throughout the study, as "predictor" may mislead readers about the nature of the research conducted. In certain instances, the term "predictor" indicates to certain readers that causal analysis is performed. But in this paper's study, "endogeneity" is obviously overlooked. Therefore, it is more suitable to avoid the term "predictor" and instead use "association" in this context.

2. I would present a missing analysis, do a sensitivity analysis using imputed values for missing data, and report on how the analysis altered (if any) as a result.

3. Clearly the PCV uptake impacts the RARI in children. Thus, in Table 2 I would add the PCV uptake variable as an independent variable in the analysis. Otherwise this will create omitted variable bias.

4. The price type of the fuel used for cooking might be used as an indication of socioeconomic class. I would advise grouping them as expensive, moderately priced, and inexpensive cooking fuels and using that variable instead of the actual cooking fuel (electricity, coal, wood etc) used.

5. In PCV analysis it would be great and much more useful if a bivariate constabulary analysis of PCV uptake and socioeconomic indicators are reported.

6. The discussion part might be enhanced by arguing in accordance with the policy context and comparing the findings to those of other nations. This would be beneficial for international readers.

Minor comments:

1. Add CU5 abbreviation

7. PLOS authors have the option to publish the peer review history of their article (what does this mean?). If published, this will include your full peer review and any attached files.

Reviewer #4: No

Reviewer #5: No

---

## [Author Response · Author response to Decision Letter 2]

15 Mar 2023

As per email correspondence we have disregarded the comments of Reviewer 4 and respond to Reviewer 5: Thanks for your additional review of our paper. We have further revised our paper accordingly. Please find our responses below each point in blue text. 

Reviewer #5: This paper used 2014 UNICEF Malawi Multiple Cluster Indicator Survey to analyse socio economic predictors for RARI un CU% and PCV uptake in children aged 1-36 months. I hope the authors find the following comments and suggestions helpful.

Major comments:

1. I would advise that the authors use the term "association" instead of "predictor" in the title and throughout the study, as "predictor" may mislead readers about the nature of the research conducted. In certain instances, the term "predictor" indicates to certain readers that causal analysis is performed. But in this paper's study, "endogeneity" is obviously overlooked. Therefore, it is more suitable to avoid the term "predictor" and instead use "association" in this context.

Authors response: Thank you, we have changed “predictor” to “association” throughout.

2. I would present a missing analysis, do a sensitivity analysis using imputed values for missing data, and report on how the analysis altered (if any) as a result.

Authors response: Please see our response to concerns about missing data in our previous revision (R2), which included the newly added (in R2) supplementary Table S1:

Authors response: thanks for this suggestion, we have now done this and added it as Table S1 in the supplementary material of our paper. This Table is the same as the first 5 columns of Table 2 of our paper, but with an added column for “missing data” on RARI, in addition to the “Yes” and “No” columns. From this, we can see that the proportion in each category of each variable missing data on RARI are similar – the proportion missing data on RARI only varies by 8 percentage points (e.g. for child age) or less across the categories of each socio-economic variable (Table S1). Therefore, our main RARI analysis (Table 2) is unlikely to be biased. We have added the following text on page 20 of our paper to explain this:

“We report socio-economic characteristics of those with missing data on RARI separately in supplementary Table S1 and find the proportion missing data on RARI only varies by 8 percentage points (e.g. for child age) or less across the categories of each socio-economic variable (Table S1), suggesting our analyses below are unlikely to be biased by this missing data.”

This sensitivity analysis strongly suggests that imputing missing data would not significantly alter our results or conclusions.

3. Clearly the PCV uptake impacts the RARI in children. Thus, in Table 2 I would add the PCV uptake variable as an independent variable in the analysis. Otherwise this will create omitted variable bias.

Authors response: As we explain in our paper, the PCV data is only available for children aged 1-36 months. Our RARI analysis is separate and for all children aged under 5 years old (0-59 months) therefore we can’t include PCV in this analysis as it would change the sample for our RARI analysis completely.

4. The price type of the fuel used for cooking might be used as an indication of socioeconomic class. I would advise grouping them as expensive, moderately priced, and inexpensive cooking fuels and using that variable instead of the actual cooking fuel (electricity, coal, wood etc) used.

Authors response: There is no data on how much the fuel costs in the dataset (MICS 2014) that we use or on how much fuel each household uses so we are unable to determine the amount spent on fuel accurately. We also already include wealth quintile in our analysis separately.

5. In PCV analysis it would be great and much more useful if a bivariate constabulary analysis of PCV uptake and socioeconomic indicators are reported.

Authors response: In Table 4 we already report results separately for each dose of PCV: PCV1, PCV2, and PCV3 by socio-economic indicators both bivariate for each variable, and multivariable. We are not familiar with “constabulary analysis” and can’t find reference to it online.

6. The discussion part might be enhanced by arguing in accordance with the policy context and comparing the findings to those of other nations. This would be beneficial for international readers.

Authors response: This study uses nationally representative data from Malawi only and is therefore highly specific to the Malawian context. We therefore do not think it is appropriate to compare with policies of other countries, and the manuscript is already very long.

Minor comments:

1. Add CU5 abbreviation

Authors response: Thank you, we have added CU5 to our list of abbreviations, and added it in brackets after “children under 5” the first time it is mentioned in the abstract.

---

## [Editor Report · Decision Letter 3]

16 Mar 2023

What is associated with reported acute respiratory infection in children under 5 and PCV vaccination in children aged 1-36 months in Malawi? A secondary data analysis using the Malawi 2014 MICS survey

PONE-D-21-21665R3

Dear Dr. Colbourn,

We’re pleased to inform you that your manuscript has been judged scientifically suitable for publication and will be formally accepted for publication once it meets all outstanding technical requirements.

Kind regards,

Shinya Tsuzuki, MD, MSc

Academic Editor

PLOS ONE
---

## [Editor Report · Acceptance letter]

23 Mar 2023

PONE-D-21-21665R3 

What is associated with reported acute respiratory infection in children under 5 and PCV vaccination in children aged 1-36 months in Malawi? A secondary data analysis using the Malawi 2014 MICS survey 

Dear Dr. Colbourn:

I'm pleased to inform you that your manuscript has been deemed suitable for publication in PLOS ONE. Congratulations! Your manuscript is now with our production department. 

Kind regards, 

on behalf of

Dr. Shinya Tsuzuki 

Academic Editor

PLOS ONE